# Mental Well-Being and Sexual Intimacy among Men and Gender Diverse People Who Have Sex with Men during the First UK COVID-19 Lockdown: A Mixed-Methods Study

**DOI:** 10.3390/ijerph19126985

**Published:** 2022-06-07

**Authors:** Natalie L. Edelman, T. Charles Witzel, Phil Samba, Will Nutland, Tom Nadarzynski

**Affiliations:** 1School of Sport & Health Sciences, University of Brighton, Brighton BN1 9PH, UK; 2Primary Care & Public Health, Brighton & Sussex Medical School, Brighton BN1 9PX, UK; 3Institute for Global Health, University College London, London NW2 2QG, UK; c.witzel@ucl.ac.uk; 4Department of Public Health, Environments and Society, London School of Hygiene and Tropical Medicine, London WC1H 9SH, UK; prep4qmoc@prepster.info (P.S.); will.nutland@lshtm.ac.uk (W.N.); 5PrEPster, The Love Tank, London E1 4AQ, UK; 6School of Social Sciences, University of Westminster, London W1B 2HW, UK; t.nadarzynski@westminster.ac.uk

**Keywords:** MSM, gender-diverse, sexual behaviour, COVID-19, mental health, well-being, intimacy, mixed-methods

## Abstract

This mixed-methods study aimed to explore mental well-being, circumstances and strategies around managing sexual intimacy and risk during the first UK COVID-19 lockdown (Spring 2020) among men and gender diverse people who have sex with men (MGDPSM), commencing while lockdown was in progress. n = 1429 MGDPSM completed the survey and 14 undertook an in-depth interview. Low mental well-being was reported by 49.6% of the survey participants. Low mental well-being was not predicted by relationship and living circumstance, sexual networking app use, or by casual sexual partners. Low mental well-being was associated with more frequent COVID-19 anxiety (OR = 5.08 CI: 3.74, 6.88 *p* < 0.001) and with younger age (18–24 years OR = 2.23 CI:1.41–3.53 *p* = 0.001, 25–34 years OR = 1.45 CI:1.04–2.02 *p* = 0.029, 35–44 years OR = 1.41 CI:1.00–1.99 *p* = 0.052). The interview participants understood their lockdown experiences as being relative to normalcy, and those experiencing more dramatic changes faced greater challenges. Living with partners was felt to protect well-being. Many participants reported intimacy interruption challenges. The findings indicate that mental well-being is predicted by age and COVID-19 impact, highlighting opportunities for targeting MGDPSM who are most vulnerable to poor mental health. Services that support MGDPSM during COVID-19 recovery efforts must provide non-judgemental and affirming support.

## 1. Introduction

The COVID-19 global pandemic has seen an unprecedented enforcement of restrictions to travel and physical contact across much of the globe, in order to reduce infections and preserve life. In the UK, the first national lockdown, announced by the UK and devolved governments, began on 23 March 2020; it restricted all non-essential travel and closed non-essential services, banning social contact outside the home and limiting exercise outside the home to once a day [1,2,3,4]. These restrictions were gradually lifted through May and June with a phased return to schools and the re-opening of non-essential shops on 15 June [5]. From then onwards, England, Wales, Northern Ireland and Scotland faced intermittent tiered restrictions (where different areas experienced a different intensity of measures) and lockdowns (where entire nations faced the same restrictions) [2,3,5].

The COVID-19 pandemic has impacted on the mental health, well-being and sexual and social lifestyles of UK, and other populations [6,7]; in the UK, this impact has been felt through both the imposition of restrictions and broader fears and insecurity associated with contracting the infection, as well as income and other factors [8]. A UK general population cross-sectional survey that was undertaken four weeks after the first UK lockdown began indicated that 52% of respondents screened positive for a common mental health disorder (CMD) [9]. Depression, anxiety, stress and other morbidities have also been reported to increase during the pandemic in several countries [7].

Men and gender diverse people who have sex with men (MGDPSM) face increased health and well-being challenges compared to the general population [10,11,12,13]. These challenges include precarious social networks and societal hostility, HIV and other sexually transmitted infections (STIs) [13,14]. Related to these challenges, MGDPSM are also more likely than many other populations to experience mental distress and CMDs [15].

Evidence from the National Survey of Sexual Attitudes and Lifestyles’ COVID-19 (Natsal-COVID) study suggests an overall decrease in the number of sexual partners in general populations during the first UK lockdown, coupled with difficulties in accessing condoms [16,17]. Natsal-COVID qualitative data point towards a variety of emotional issues even among those in established partnerships [18]. The links between mental well-being, intimacy and sexual risk are complex, perhaps especially in times of increased precarity and vulnerability such as during the COVID-19 pandemic. A recent study from the Republic of Ireland found that 75% of MSM reported that their mental health worsened during the COVID-19 pandemic, and that the closure of LGBT venues and limits on socialising led to decreases in well-being [19]. Other studies have examined the sexual behaviour of LGBT people during the COVID-19 government restrictions, although these largely do not focus on mental well-being [20,21,22,23]. There is a critical need to understand the interactions between these areas in order to plan services during COVID-19 recovery efforts and waves of future variants or other zoonotic disease that adequately address the unique experiences of MGDPSM. This study aimed to explore the mental well-being, circumstances and strategies around managing sexual intimacy during the first UK COVID-19 lockdown among MGDPSM.

## 2. Materials and Methods

### 2.1. Design

A mixed-methods design was used to observe the overall trends in the experiences of MGDPSM and in order to gain richer insights into those experiences. Qualitative data was triangulated with the quantitative data by explaining and contrasting with observed statistical associations. This involved an approach termed ‘following the thread’, whereby qualitative data are used to add depth and nuance to a quantitative dataset [24]. A cross-sectional anonymous online survey was conducted amongst MGDPSM from 20 April–25 May 2020 with a qualitative component exploring the emerging findings between June 2020 and January 2021.

### 2.2. Inclusion and Exclusion Criteria

The study was open to cis and trans men, trans women and non-gender conforming people who have sex with cis and trans men that were aged 18 years or over and residing in the UK at the time of completion. Screening questions at the beginning of the questionnaire were used to exclude those who did not meet the criteria. As the survey was anonymous and not linked to those who volunteered for the qualitative research, demographic questions including age, sexual orientation and gender identity were asked when responding to participant enquiries about undertaking an interview. We excluded from the study cis-women who were not gender-non-conforming, and respondents who reported having sex exclusively with women.

### 2.3. Survey Recruitment and Data Collection

The study advert was circulated on Twitter, Facebook, Instagram and Grindr using a variety of MGDPSM hashtags and offering participants the chance to enter a prize draw worth GBP 75. By clicking a link in the advert, participants were taken to a webpage containing participant information and study consent. Completion of this then allowed the participant to proceed to the anonymous 40-item questionnaire which included items on mental well-being (using the seven-item Short Warwick-Edinburgh Mental Well-being Scale (SWEMWBS); use of sexual networking apps; sexual experiences; living circumstances; and relationship status, and uptake of sexual health interventions such as STI testing and PrEP (please see Appendix A for measures used). At the end of the questionnaire, the participants were provided with information about the qualitative interview study and invited to provide their name and contact details on a separate page if they were interested in participating. This information was kept separately from the survey data in order to preserve survey anonymity.

### 2.4. Statistical Analysis

The survey took a convenience sample approach with the aim of recruiting as many MGDPSM as possible. For this reason, a sample size calculation was not conducted *a priori*. Instead, an events-per-variable approach was used to avoid entering too many exposures into the model for the available sample size once the data had been cleaned, using the 10 extra events per exposure rule-of-thumb [25]. Data were uploaded from Qualtrics into SPSS (v25) for analysis. An available case analysis was used (conducted by NE), excluding cases for which there was missing data for any of the variables that were included in the analysis. The raw scores for the SWEMWBS were transformed to metric scores using a tabular guide that provides a metric equivalent for every possible raw score, in line with recommended use. For this analysis, a multivariable model was developed to identify independent predictors of mental well-being. To avoid over-fitting the model, bivariate analyses were not used to determine which variables to include. In keeping with the exploratory nature of the analysis, variables were selected for model entry if they were theorised to predict mental well-being. Variables that were anticipated to predict mental well-being were entered into the model, representing: age group; relationship and living status; number of casual partners during lockdown; sustained or increased number of partners during lockdown; sustained or increased number of non-physical sexual contacts during lockdown; frequency of sexual network app opening; change in time spent just chatting online; and anxiety about the pandemic. Reference categories were chosen to represent the status hypothesised to be indicative of higher well-being and not necessarily based on magnitude, e.g., for the variable ‘frequency of app opening’, the chosen reference category was ‘several times a week’ rather than ‘once a week or less’. In order to generate odds ratios and align mental well-being with depression we dichotomised the SWEMBS scores to ‘average or higher than average well-being’ or ‘poor or very poor well-being’, with the latter being indicative of possible or probable depression based on previous research [26].

### 2.5. In-Depth Interview Recruitment and Data Collection

Survey participants who had provided contact details for the qualitative interview study were contacted via email to discuss participation, provide a participant information sheet and to schedule a time for interview. They were asked for basic demographic information in their reply email. Participants were purposively selected to include a range of ethnicities, sexual orientations and gender identities.

In-depth interviews were between 40 min and 1 h long and followed a topic guide that was designed to add depth to some of the key areas that emerged from the cross-sectional survey. The topic guide (see Appendix A) focused on participants’ experiences of lockdown, including living arrangements, impact on mental health and anxiety, app use, sexual behaviour and changes during periods of social distancing. The analysis that is reported on in this paper focuses on those topics that are related to mental well-being and intimacy. A further section explored participants’ experiences of sexual health service access during the pandemic.

Participants provided verbal recorded consent. The interviews were conducted using videotelephony and audio-recorded before being transcribed verbatim.

### 2.6. Qualitative Analysis

The analysis drew on principles from narrative and framework approaches in order to examine how personal understandings of impacts compared with those of peers [27,28,29]. A broad deductive framework was first developed with the key areas of enquiry for the qualitative analysis. All qualitative data were then coded to these key areas. All participant accounts were inductively interpreted in the context of their interviews and in comparison with others, creating narratives that represented broad experiences with negative cases used to explore the divergence and meanings behind this. During the analysis, special attention was paid to areas of interest which emerged in the survey in order to provide depth and nuance. One researcher (TCW) performed the analysis. The full analysis was presented to and discussed with the team to ensure it reflected a collective understanding of participant narratives and experiences. Theoretical saturation was not assessed; rather, we sought to understand a range of experiences and how they compared across cases. We report the age ranges for qualitative participants to ensure confidentiality.

### 2.7. Ethics Approval

Ethical approval to conduct the study was granted by the University of Westminster Research Governance and Ethics Committee (reference: ETH1920-1601) and the London School of Hygiene and Tropical Medicine Observational Research Ethics committee (reference: 22421).

## 3. Results

### 3.1. Survey Findings

Of n = 1429 participants in the total database, 82.0% (n = 1172) were included in this available case analysis. Among the n = 1172 included, 11.6% (n = 136) reported one casual sex partner during lockdown and 11.3% (n = 132) reported two or more. A total of 25.9% (n = 303) reported feeling anxious about COVID-19 more than half the time and 49.6% (n = 581) had a mental well-being score that was consistent with possible or probable depression. A total of 28.6% (n = 335) reported living alone, and the median age of participants was 35 years (min–max = 18−70 years) with 14.4% (n = 169) aged 18–24 years. The sample was mostly of White-UK origin (n = 1004, 85.7%) and 65.8% (n = 770) reported degree level qualification or higher. A total of 92.6% (n = 1085) reported being sexually interested in men only. Among the n = 1172 participants that were included in this analysis, n = 1139 (97.2%) identified as male, including n = 14 (1.3%) who reported female sex at birth. Of the n = 1125 who reported male sex at birth, n = 4 (0.36%) identified as female or trans female. A further n = 29 (2.5%) reported their gender identity as non-binary or other.

Among the 18.0% (n = 257) who were excluded, 86.8% (n = 223) were removed because they did not provide responses to one or more of the seven items that constituted the well-being scale. Participants that were excluded from the analysis were less likely to report White ethnicity −79.0% (n = 203), Chi-Square = 7.341 *p* = 0.007, and to report education to degree level 59.5% (n = 153), Chi-Square = 3.57 *p* = 0.059. There was no difference in age or sexual orientation between those that were included and excluded using inferential statistics. Owing to small frequencies, differences in gender identity could not be assessed in this way but were similar among those excluded: 96.1% (n = 247) identified as male, including n = 3 (1.2%) who reported female sex at birth. Of the n = 224 who reported male sex at birth, n = 2 (0.9%) identified as female or trans female. Of all excluded cases, a further n = 8 (3.11%) reported their gender identity as non-binary or other.

Due to non-normality of the residuals, robust standard errors were computed and are presented in Table 1, which illustrates the multivariable regression model depicting predictors of mental well-being.

Overall, mental well-being scores that were consistent with possible or probable depression were reported by 49.6% of the sample. A linear association between age group and well-being was observed; those aged 18–24 years reported the highest percentage of mental well-being that was indicative of possible-probable depression (62.7%) and those aged 45+ years reported the lowest percentage (40.4%). The percentage that reported being single and living alone increased with age, from 8.9% among 18–24-year-olds to 41.1% among those aged 45+. COVID-19 anxiety was negatively associated with well-being such that lower well-being that was indicative of possible or probable depression was associated with being anxious about COVID-19 more than half the time. Networking app use, non-physical sexual contact (defined in the questionnaire as sexting, webcam or phone sex, exchange of naked pictures etc.), and physical sexual contact with casual partners were not predictive of well-being. The majority of respondents (78.9%) reported a reduction in casual partners during lockdown and 77.1% reported no casual sex partners at all during lockdown. Overall, the model explained only a small amount of the variance in well-being (R-squared = 0.096).

### 3.2. Interview Findings

Fourteen participants took part in in-depth interviews (13 cisgender MSM and one trans MSM). A narrative framework analysis revealed three primary narratives around mental well-being, living circumstances and social/sexual intimacy: descriptions relative to ‘normal’; coping, isolation and mental health; and managing intimacy deficits.

#### 3.2.1. Descriptions Relative to ‘Normal’

All narratives in the interviews focused on descriptions of individuals’ lives relative to their ‘normal’ pre-pandemic states, a framing which was consistently drawn upon to interpret their experiences throughout the COVID-19 crisis.

*I used to not work so much from home. I did do a decent amount of work from home, but I would always be in an office a couple of days a week. Being Civil Service, they’ve tried to keep us away from offices and spreading germs that we might have. So it’s been very much at home for me. But really, I’ve got to be honest, because I work so much from home anyway, I’ve not actually had a great deal of change to myself, apart from the fact that maybe the last time I went out of the town was, I don’t know, six months ago*.(35–44 year-old gay White cisgender man)

Several participants described living with partners as providing a sense of stability and grounding that contributed to emotional resilience. These were primarily sexual partners, but in one case was a heterosexual man with whom he had a relationship akin to marriage. Although there were frustrations with increased time spent together, overall co-habituating relationships were felt to be helpful:

*During actual lockdown my partner was furloughed, so he was always at home. And that was different because I think, when you’ve been in a relationship for ten years, you need to sometimes get your own space. But, obviously, it was good to have him around a lot of the time, but I just felt a little bit claustrophobic sometimes, just because we’re used to working different shifts and getting time apart. So that was the only difference to living in lockdown*.(18–24-year-old gay Asian cisgender man)

For others, however, the change brought about by the first lockdown was dramatic and their entire way of living shifted within a very brief period, creating a great deal of uncertainty and change. This was often most pronounced amongst those who lived alone or with non-partners, and those with pre-existing mental health conditions. One participant described a loss of sense of purpose:

*Obviously, because as well as go out and socialise and see my acquaintances and all that stuff, and they’re key for me to keep socially active and feel like I’m doing something productive with my time. And so inherently because of lockdown those two things that are primarily my, I call them main kind of purpose if that makes sense. It gives me, it gives me a something to live for,* [and] *they were taken away quite quickly*.(35–44 year-old gay White cisgender man)

Although these narratives are highly intuitive, they were the cornerstone of interpreting later experiences and changes in well-being; those whose lives changed the least tended to describe positive well-being while those whose lives changed the most faced substantial challenges in coping.

#### 3.2.2. Coping, Isolation and Mental Health

All narratives described difficulties with isolation, with some coping poorly. In addition, pre-existing mental health conditions were exacerbated. Technological approaches to socialising were viewed with some ambiguity; although they were appreciated, their novelty wore off. Narratives sometimes described social networks fracturing because of isolation and one participant experienced a serious mental health crisis.

Most narratives described some degree of boredom associated with lockdown and the COVID-19 crisis. This was typically drawn out and most pronounced in those that were placed on the government’s furlough scheme.

*I wouldn’t say it was stressful. It was more boring than anything. I don’t think I had any other stronger emotions towards it. I just found it very dull because I’m quite a social, outgoing person. And to not be able to do that was very, very boring*.(25–34- year-old Asian gay cisgender man)

Technological approaches to maintaining social contact were extremely common. For some individuals this meant regular socialising with friends through video conferencing technology, while for others it was an increased reliance on social media platforms and gay focused hook-up apps. Although social media was sometimes described as helpful for maintaining connection and coping with isolation, for others it contributed to an overall sense of despair.


**
*And you mentioned that it had a mental health impact, can you sort of describe what that what that has been for you?*
**


*Just kind of feelings of despair is too big a word, but just kind of like, you just kind of look at the whole situation and the kind of the whole doom scrolling kind of aspect of like social media on Twitter and stuff. And it’s just constantly bad news.* [...] *Like, I like Twitter, I use it a lot in some ways, like it helps with mental health.* […] *And just* [with] *COVID and kind of being stuck inside at the same time. Certainly* [it] *kind of exacerbated those feelings*.(25–34-year-old gay White cisgender man)

Some participants who lived alone reported fatigue with using technological approaches to manage their feelings of isolation. These narratives sometimes described the fracturing of social networks during the pandemic because of their reluctance to engage with online socialising. This was especially problematic for those who had more precarious social networks because of familial difficulties related to sexual identity, as well as for those who were socially isolated for other reasons.

*I actually felt guilt that I wasn’t socialising virtually because, particularly early on* […] *everybody was doing a pub quiz every weekend and it’s almost there was just too much going on. And I felt a bit of guilt from friends that I wasn’t getting involved virtually.* […] *It sort of compounded the loneliness a bit more because I felt like I was letting my friends down by not getting involved digitally. And as people have got used to not having me around, I feel like now, as things have opened up, I don’t really feel like there are opportunities nor the desire from people who I used to hang out with to hang out again because they’ve got used to not hearing from me*.(25–34-year-old gay White cisgender man)

One participant with a pre-existing mental health condition experienced a catastrophic deterioration in his mental health in the early stages of the first lockdown and attempted suicide. This was attributed to a substantial disruption to his coping strategies and suddenly being extremely socially isolated as he lived alone.

#### 3.2.3. Managing Intimacy Deficits and ‘Letting-Go’

Intimacy, and a lack there-of, featured substantially in participant narratives. Interrupting intimacy was difficult for many, especially participants who lived on their own or with family/flat mates, rather than with sexual partners. This did not affect individuals uniformly; although it was extremely difficult for some, especially those for whom sexual activity was a core component of their sexual identity, others found it relatively easy to halt sexual activity for a period of time. These feelings were not static, and there were substantial changes throughout various stages of lockdown described.

For some, the initial limits placed on sex themselves increased feelings of frustration, leading to growing sexual desire:

*So, the feelings imposed were a little bit frustration. I think my level of horniness type thing was a lot higher because you couldn’t have it. So then you were a bit more on edge and you were a bit like everything would set you off*.(19–24-year-old Asian gay cisgender man)

Generally, the narratives contained a substantial amount of ambivalence about the impact of government restrictions on intimacy; restrictions were acknowledged as important, with a strong emphasis placed on abstaining as an act of good gay citizenship to protect the most vulnerable in society. The halting of sexual behaviour was also sometimes regarded as an opportunity to slow down and take stock of life more broadly, especially for those whose identity prior to the COVID-19 pandemic emphasised sex as an important aspect of personhood.

*I think it’s double edged. On the one hand, it’s been quite lonely and it’s actually frustrating. On the other hand, it’s been a time of reflection and thinking about what I want and thinking about intimacies that have been good and not so good. So, in a way that’s been useful,* [but if] *it carries on and on, and on, and on, then I think there’ll be more frustration*.(25–34-year-old Gay Asian trans man)

Many resorted to technological solutions, using hook-up apps and other instant messaging as a replacement for sex in person, and to maintain social contact with sexual networks, although this was not felt to be a sustainable alternative in the long-term.

When participants described re-initiating sexual behaviour following initial halting, this typically included an acknowledgment of their own complex feelings that was related to a wider stigma around sexual behaviour with partners outside the household during this period, and then ‘letting go’ of guilt and shame. Participants who re-initiated sexual activity earlier in the period government restrictions were those who lived alone or with housemates, rather than friends or partners. Many who re-engaged in sexual activity described acknowledging and submitting to sexual desire, while for others’ narratives it focused on continuing with life in the face of the pandemic.

*And I think it put physical limitations on the mental limitations from both my perspective, and others, and. But then I think, I think, as time went on people’s mentality. And my own mentality shifted- change from I need to protect myself to, I need to live*.(35–44 year-old, White, gay cisgender man)

It was acknowledged that letting go of self-stigma around sexual activity was challenging and that rebounding feelings of guilt and shame could sometimes occur. One cisgender man described a week in which he was highly sexually active before a new set of government restrictions came in, and the emotional impact:

*I think just before the new restrictions* [of the second lockdown]*, there was a crazy week when there was a sauna that was still open. I went to it twice in one week and I saw three guys in one week and I was exhausted, and I just reflected afterwards. That comes from anxiety, that that and it came I think, from sadness, I think it came from isolation. When I thought about it afterwards, I felt quite shit afterwards, after that week*.(25–34 year-old Gay Asian trans man)

This indicates that ‘letting go’ of the stigma around sexual activity is perhaps not a straightforward process and involves complex emotional responses and the potential for re-bounding guilt and shame.

## 4. Discussion

Overall, a high prevalence of poor mental well-being was reported. The survey analysis indicated that well-being was lower among young MSMGDP and those experiencing more persistent COVID-19 anxiety. Sexual behaviour (both physical and non-physical) was not indicative of well-being using multivariable modelling, nor was the use of sexual networking apps for both sexual and non-sexual purposes. Overall, the variance in the model was not well explained by the investigated exposures, suggesting that well-being may be explained by other variables that were not captured in the questionnaire, such as loneliness, access to social support networks and extent of general life changes (a key qualitative finding—see next paragraph).

The qualitative analysis of participants’ narratives reveals a complex picture. Well-being during this period was highly contingent on the extent to which individuals’ lives had changed during government restrictions, and on their living situations. Substantial impacts on well-being were observed, ranging from boredom to anxiety, and to suicidality for one. Technological approaches to maintaining social contact with social and sexual networks were very common but viewed with some ambivalence, as they were not seen as a viable alternative in the long term. Government restrictions due to the COVID-19 crisis led to the fracturing of weaker social ties for some, especially those who lived alone and/or described themselves as introverts.

Although the interruption of sexual behaviour affected participants in qualitative research differently, none found it to be entirely straightforward. Halting or limiting sexual activity was felt to be an important component of good gay citizenship, in line with other research [30]. Participants universally acknowledged that these reductions were not viable in the long-term. While those who re-initiated sexual activity described letting go of stigma, this was not necessarily durable and led to complex emotional responses.

The survey analysis revealed no association between living circumstances and mental well-being, contrasting with qualitative findings that living with others had a protective effect. This discrepancy may reflect sample bias in the qualitative study towards those with close relationships with those they lived with. The relationship between mental well-being and living circumstances is perhaps contingent on the *quality* of relationships—a variable that was not captured in the survey questionnaire.

### 4.1. Comparison to Other Studies

In line with existing research in general populations, our results show that large proportions (49.6%) experienced low mental well-being [31], and that these potential impacts were associated with age [32]. In contrast with a study in Romania, where those who were most likely to have anxiety were between 18–24 and 55–65 [32], in our sample low mental well-being was less common in older age groups.

Qualitative findings indicated that living with others, particularly sexual partners, might be protective of mental well-being. Other previous research also indicates that living with a partner can be protective of mental health for LGBT people [12]. Baseline data (captured March–April 2020) for the UK-based general population COVID-19 Psychological Well-being Survey indicated similarly that living with more people was protective of loneliness [33]. Qualitative findings from the Natsal-COVID study indicated that those seeking sex outside of their household wrestled with similar complexities regarding judgement, risk and mental health when making those decisions [34].

The high prevalence of low mental well-being (corresponding to possible or probable depression) compares with UK general population data indicating the April 2020 Common Mental Disorder prevalence as 37.2% [8] and 52.0% [9]. Lower well-being among participants aged 18–24 years is in line with existing research on the needs of sexual and gender minority groups [12,35,36]. Online qualitative research with LGBTQI youth revealed how the COVID-19 lockdown had a negative effect on mental health in ways both common to all young people, but also specific to gender and sexual identity—such as the expression of such identities being relegated to online environments [37]. The UK-based general population COVID-19 Psychological Well-being Survey also found that the psychological well-being of young people was particularly impacted by the pandemic [33]. Thus, the lower mental well-being reported by younger participants may reflect intersections of age-related, and gender and sexual minority related stresses that were exacerbated by the fracturing of social networks during the COVID-19 pandemic. This finding is important in the context of UK evidence indicating greater difficulties accessing sexual health services among young men, than among other gender and age groups during the pandemic [38]. Further analysis of the questionnaire items that were related to access to sexual health services and interventions such as STI testing and PrEP will be presented in a further publication.

Among the sample in this analysis, only a small proportion reported one or more casual sex partner during lockdown n = 136 (11.6%), with a similar number reporting two or more—n = 132 (11.3%). Although no directly comparable data are available from other studies, the Natsal-COVID study reported a similar overall frequency for Intimate Partner Contact outside the Household of 9.9% (95%CI 9.1–10.6%), occurring more commonly among gay 9·5% (15·3–24·6) and bisexual 16·9% (13·3–21·1) populations [39].

### 4.2. Limitations and Strengths

The online recruitment approach may limit the generalisability of the findings to UK MSMGDP overall, introducing a bias towards digitally-enfranchised populations. A bias was observed in the quantitative data set—those excluded from the study due to non-completion of well-being and/or other items were more likely to report non-White ethnicity and an education below degree level. This may impact the generalisability of the findings, although there were no observable differences in age, gender identity or sexual orientation. Loneliness and the quality of relationships were not captured quantitatively in our study and may have contributed considerably to the unexplained variance in the model. The short-form version of the Edinburgh-Warwick scale focuses on functional aspects of mental well-being rather than affect; the longer-form may have provided a more complete picture of how mental well-being is associated with psychosocial and sexual well-being for this population (but may also have negatively impacted on the completion rates).

The quantitative component of this study captured a unique point in time; findings may not be transferable to the experiences of MGDPSM in subsequent lockdowns. The analysis of general population data pre-COVID-19 and during the last year indicates that April saw a peak of CMDs, with rates then declining by July 2020 [8]. The quantitative data were gathered during the first lockdown, the only sexual health survey of this population to capture survey data in real-time rather than retrospectively. Participants were also able to complete the survey at any time during lockdown, such that some participants would have experienced more time under lockdown measures than others—this variable was not captured. The qualitative data were captured over a longer timeframe and thus were more susceptible to recall bias, but they do provide a rich understanding of experiences as the COVID-19 crisis progressed in the UK, and the experiences of MGDPSM during subsequent lockdowns.

Due to low frequencies, it was not possible to disaggregate the survey responses for participants in monogamous relationships who were living alone versus those living with others. It was also not possible to disaggregate MSM, gender diverse and trans participants, such that differences in their experiences were not captured in the quantitative analysis. Nonetheless, a strength of this study was that it aimed to explore the behaviours of trans and cis men as well as trans women and non-binary people who have sex with other men. These included men who identify as gay, bisexual and queer. We did not gather data in order to enable the analysis of a reduction in casual partners that was specific to those who had casual partners prior to lockdown.

Due to recruitment challenges, we were only able to interview MSM (cis and trans) rather than trans women or non-binary people. The qualitative findings are therefore reflective of the experiences of MSM (gay, bisexual and queer) rather than MGDPSM as a whole. Nonetheless, our interviews provide in-depth insight into the interactions between broader psychosocial factors with sexual behaviour and well-being.

## 5. Conclusions

Our results shed light onto potential areas for intervention that should be considered during recovery from the COVID-19 crisis. Based on the survey data, younger MGDPSM may have an increased need for targeted intervention to support both their sexual health and mental well-being.

Our study as a whole illuminates the ways in which those MGDPSM who had more fragile social networks at the outset of the pandemic are likely to face increased hardship and may require targeted intervention. This is also true of those with pre-existing mental health conditions, both diagnosed and undiagnosed, and those who have spent much of the COVID-19 crisis living on their own or in accommodation shared with non-partners and non-family members.

Both mental and sexual health services serving MGDPSM must be attentive to their unique circumstances during the COVID-19 crisis. This includes providing non-judgemental and affirming services, particularly when discussing sensitive topics such as sexual activity during periods of government restrictions. Providing such services is vital in engaging a range of MGDPSM in order to support their sexual and mental well-being.

More research is needed to understand how the social and economic position of MGDPSM may impact on their well-being, identifying particularly vulnerable groups in need of intervention [8,12]. This speaks to broader calls for the greater consideration of intersectionality in both general and LGBTQI specific research [40]. The bias in the completion of mental well-being items towards White and highly educated participants also indicates that further work is needed to improve inclusivity regarding not only overall study participation, but the acceptability of specific questionnaire scales and individual items.

## Figures and Tables

**Table 1 ijerph-19-06985-t001:** Predictors of low mental well-being during the UK lockdown (March–April 2020) in men and gender diverse people who have sex with men.

Variable	N (Valid %)	Adjusted OR ¥	Robust SE *	*p* > [Z]	[95% CI] ¥
**Age in years**					
18–24	169 (14.4)	2.23	0.52	0.01	[1.41–3.53]
25–34	410 (35.0)	1.45	0.25	0.03	[1.04–2.02]
35–44	296 (25.3)	1.41	0.25	0.05	[1.00–2.00]
45+	297 (25.3)	Ref	-		
**Relationship & living arrangement**					
Single & living alone	335 (28.6)	1.01	0.27	0.98	[0.59–1.72]
Single & living *w/o* **	481 (41.0)	0.88	0.23	0.62	[0.52–1.48]
Open/comp. rel. & living alone or *w/o* **	120 (10.2)	0.95	0.29	0.86	[0.52–1.72]
Open/comp. rel. & living with ptn	162 (13.8)	0.75	0.23	0.34	[0.41–1.36]
Monogamous rel. living alone or *w/o* **	74 (6.3)	Ref			
**Frequency of COVID-19 anxiety**					
More than half the time	303 (25.9)	5.08	0.79	0.01	[3.74–6.88]
Half the time or less	869 (74.1)	Ref			
**Number of casual partners**					
1–2 casual partners	136 (11.6)	1.01	0.18	0.94	[0.72–1.43]
3 casual partners	132 (11.3)	1.16	0.29	0.55	[0.71–1.89]
0 casual partners	904 (77.1)	Ref			
**Change in number of casual partners**					
Same or more	247 (21.1)	1.03	0.18	0.88	[0.74–1.43]
Less than usual	925 (78.9)	Ref			
**Frequency of sexual network app opening**					
Several times a day	543 (46.3)	1.15	0.20	0.43	[0.82–1.61]
Every day	229 (19.5) 164 (14.0)	1.01	0.20	0.97	[0.68–1.50]
Once a week or less	236 (20.1)	1.06	0.23	0.79	[0.69–1.63]
Several times a week		Ref			
**Change in time spent chatting online**	298 (25.4)				
Reduced	277 (23.6)	1.03	0.18	0.86	[0.73–1.45]
Stayed the same	597 (50.9)	0.91	0.15	0.53	[0.65–1.24]
Increased		Ref			
**Change in non-physical sex activity**	280 (23.9)				
Reduced	488 (41.6)	0.89	0.15	0.54	[0.64–1.26]
Increased	404 (34.5)	1.01	0.16	0.90	[0.75–1.38]
Stayed the same		Ref			

Note: * Standard Error ****** With Others ¥ Odds Ratio adjusted for all other variables in the model.

## Data Availability

The data presented in this study are available on request from the Chief Investigator, Tom Nadarzynski. The data are not publicly available due to their highly sensitive nature, in line with the data management and storage plan approved by the University of Westminster.

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
