# Peer review of "Mental Well-Being and Sexual Intimacy among Men and Gender Diverse People Who Have Sex with Men during the First UK COVID-19 Lockdown: A Mixed-Methods Study"

_ijerph, 2022, doi:10.3390/ijerph19126985_

Round 1
Reviewer 1 Report
Dear authors,
Congratualions for your work. Mixed studies are always complex to conduct, where triangulation of results is a challenge.
Despite the merit of your work, I would like to see more information in your article about quantitative methodology and qualitative methodology (e.g., type of questions).
I have a some issues:
Abstract:
Line 14 - Spring 2000 (2020?)
2. Materials and Methods
Can you provide and describe of instruments and type of questions?
2.6. Qualitative Analysis
There were independent judges in the analysis of the qualitative data?
3.2. Interview findings
Well... In this section I don't know what/what type of questions were designed. For example, you say "All narratives in interviews focused on descriptions of individual´s lives..." but you only transcribe one interview. Can you provide a short parts os this experience of anothers interviews?
The authors says "Several participants described..." How many? 13 interviews? 12, 11...? In fact, the authors only describe one interview with example in all papper. The readers need more information about this experience.
Can the authors provide, for example, cloud words for qualitative research and a final model with triangulating the two methodology?
The discussion and conclusion must be improved.
Regards,
Author Response
Dear Editor and reviewers,
The authors thank you for a constructive and speedy review process. We have now made amendments as suggested to our manuscript and outline the changes below.
We hope these responses are sufficient and hope that this manuscript can be published in due course.
Best wishes
Dr Natalie L Edelman
Dr T Charles Witzel

Reviewer 2 Report
The present paper investigated the mental well-being and associated factors during the first UK COVID-19 lockdown among Men and Gender Diverse People who have Sex with Men (MGDPSM). The issue is relevant as most research on the COVID-19 pandemic impact on mental wellbeing has mainly focused on general populations the examination of impact on sexual minorities has received less attention. The authors employing a mixed methods approach and the focus on sexual minorities certainly represents a strong point of the paper. However, there a number of issues that needs to be addressed before the paper is considered as deserving publication.
Abstract
The abstract is rather long and the opening line is redundant. It can be cut
Methods and Materials
A section of Measures is missing. In lines 66-69 authors state that they used a “40-item questionnaire which included items on mental well-being (using the seven-item Shortened Warwick-Edinburgh Mental Well-being Scale (SWEMWBS), sexual networking app use, sexual experiences, living circumstances and relationship status, and uptake of sexual health interventions such as STI testing and PrEP- please see Appendix A”
No information is provided in Appendix a for either the SWEMES scale nor the uptake o sexual health interventions. I suggest the authors add a Measures section where they provide detailed information on the survey items and the mental wellbeing scale including sample items and scoring method.
Statistical Analysis
Please provide the type of software used for performing analysis.
In line 113 authors state that “Raw scores for the SWEMWBS were transformed to metric scores in line with recommended use” but they provide no reference as to this and no information on how scores were calculated.
In lines 126-127 authors state “ In order to generate odds ratios and align EW with depression we dichotomised the SWEMBS scores to ‘average or higher than average well-being’ or ‘poor or very poor well-being’, with the latter being indicative of possible or probable depression.” Again, authors did not provide any information about the scale nor its scoring method and cutoffs indicating clinical relevance with regard to depression.
Results
In lines 196-197 authors write "Overall, EW scores consistent with possible or probable depression were reported by 49.6% of the sample. " In line with the comments above, it is not clear how EW scores are aligned with probable depression.
Discussion
In the opening paragraph authors state that well-being was lower among young MSMGDP which is in line with a large body of research from general population studies worldwide suggesting the young as particularly vulnerable to pandemic distress and low mental wellbeing. I think it would be important for authors to place their work withing this growing literature on pandemic impact on general population (see for instance https://doi.org/10.3390/healthcare10020247; https://doi.org/10.3390/ijerph17114151 ). I believe this would highly increase the value of the paper as it would help highlight commonalities and differences between reactions of specific population groups like sexual minorities which might be particularly vulnerable in the context of a pandemic.
In lines 398-403 authors state “Thus, the lower EW reported by younger participants likely reflects the intersection of age-related and gender and sexual minority- related stresses. This finding is important in the context of UK evidence indicating greater difficulties accessing sexual health service access among young men than other gender and age groups during the pandemic [35]. Further analysis of questionnaire items related to access to sexual health services and interventions such as STI testing and PrEP will be presented in a further publication.” I found no evidence presented in this paper to support this idea. I would eliminate this paragraph altogether.
Conclusions
Authors open the conclusions section with the statement “Our results shed light onto some key interventions that should be considered during recovery from the COVID-19 crisis. Based on the survey data, younger MGDPSM may be in particular need of targeted intervention to support both their sexual health and mental well-being.” I find this a rather strong statement and a bit of a stretch. I suggest authors to reframe.
Minor Comments
In relation to comments regarding measures, the main dependent variable in this study is Emotional wellbeing (EW) which is used with the Shortened Warwick-Edinburgh Mental Well-being Scale (SWEMWBS). If the given scale, as suggested by the name, measured mental wellbeing, why then authors state Emotional wellbeing is measured throughout the MNS? I suggest changing this term with “mental wellbeing” throughout the paper and in the title.
I suggest the tittle be changed into “Emotional well-being among Men and 2 Gender Diverse People who have Sex with Men during the first 3 UK COVID-19 lockdown: a mixed-methods study” considering that variables tapping on sexual intimacy did not predict emotional wellbeing.
Author Response

(The authors gave the same response as above.)
